Multi-FusNet: fusion mapping of features for fine-grained image retrieval networks

Cui Xiaohui 1 2
Li Huan 1 2
Liu Lei 1 2
Wang Sheng 1 2
Xu Fu xufu@bjfu.edu.cn 1 2 3
1 Engineering Research Center for Forestry-oriented Intelligent Information Processing of National Forestry and Grassland Administration , Beijing , China
2 School of Information Science and Technology, Beijing Forestry University , Beijing , China
3 State Key Laboratory of Efficient Production of Forest Resources , Beijing , China
Sajid Ullah Syed
Electronic publication date: 2024 Jun 24
Publication date: 2024
Volume: 10
Electronic Location ID: e2025
Received 2023 Nov 21; Accepted 2024 Apr 8
Copyright: ©2024 Cui et al.
Copyright year: 2024
Copyright holder: Cui et al.
License: This is an open access article distributed under the terms of the Creative Commons Attribution License, which permits unrestricted use, distribution, reproduction and adaptation in any medium and for any purpose provided that it is properly attributed. For attribution, the original author(s), title, publication source (PeerJ Computer Science) and either DOI or URL of the article must be cited.
License URL: https://creativecommons.org/licenses/by/4.0/

Keywords: Fine-grained hashing, Deep hashing, Feature fusion, Attention mechanism

Funding: The National Key R&D Program of China 2022YFF1302700 The Emergency Open Competition Project of National Forestry and Grassland Administration 202303 Outstanding Youth Team Project of Central Universities QNTD202308 This research was jointly funded by the National Key R&D Program of China (2022YFF1302700), the Emergency Open Competition Project of National Forestry and Grassland Administration (202303) and Outstanding Youth Team Project of Central Universities (QNTD202308). The funders had no role in study design, data collection and analysis, decision to publish, or preparation of the manuscript.

==============================
As the diversity and volume of images continue to grow, the demand for efficient fine-grained image retrieval has surged across numerous fields. However, the current deep learning-based approaches to fine-grained image retrieval often concentrate solely on the top-layer features, neglecting the relevant information carried in the middle layer, even though these information contains more fine-grained identification content. Moreover, these methods typically employ a uniform weighting strategy during hash code mapping, risking the loss of critical region mapping—an irreversible detriment to fine-grained retrieval tasks. To address the above problems, we propose a novel method for fine-grained image retrieval that leverage feature fusion and hash mapping techniques. Our approach harnesses a multi-level feature cascade, emphasizing not just top-layer but also intermediate-layer image features, and integrates a feature fusion module at each level to enhance the extraction of discriminative information. In addition, we introduce an agent self-attention architecture, marking its first application in this context, which steers the model to prioritize on long-range features, further avoiding the loss of critical regions of the mapping. Finally, our proposed model significantly outperforms existing state-of-the-art, improving the retrieval accuracy by an average of 40% for the 12-bit dataset, 22% for the 24-bit dataset, 16% for the 32-bit dataset, and 11% for the 48-bit dataset across five publicly available fine-grained datasets. We also validate the generalization ability and performance stability of our proposed method by another five datasets and statistical significance tests. Our code can be downloaded from https://github.com/BJFU-CS2012/MuiltNet.git.

Introduction

In the big data area characterized by the rapid advancement of the Internet and the widespread use of multimedia devices, all data categories are experiencing a trend of rapid growth. With the expansion of image content categories, the significance of the application of retrieval techniques is also increasing recognized. By virtue of its ability to quickly and accurately query an image database for content similar to the input image, image retrieval, with its capacity for swift and accurate searches of image databases for content resembling a given input image, has emerged as a critical area of research in computer science. In the early days, based on the advantages of simple extraction, intuitive representation, and strong robustness, traditional image retrieval methods heavily relied on manual feature mining (Haralick, Shanmugam & Dinstein, 1973; Tico et al., 2001; Jeong, Won & Gray, 2004; Xi, Guang & Shunli, 2012), specifically including the use of edge histograms, color histograms, and wavelet texture, etc., to generate a special descriptor of the image with the help of extracted information such as color, texture, and edges, which is used as the basis for retrieval. However, these features, such as color and texture, can be significantly influenced by environmental conditions like angle and lighting, leading to poor generalizability. Therefore, researchers have proposed to use more sophisticated and fine-grained features to process complex images. Techniques like scale-invariant feature transformation (SIFT) descriptors and accelerated robust features have been adopted in image retrieval. In 2015 the concept of fine-grained image retrieval is introduced, demonstrating the application of manual features in fine-grained tasks using the SIFT method (Xie et al., 2015a). Deep learning technology, which overcomes the limitations of manual features in capturing high-level semantic information and is less sensitive to perspective and scale changes, has become prevalent for extracting image features for retrieval tasks.

Deep learning stand apart from traditional methods of manually extracting underlying image features by its ability to autonomously learn and extract high-level semantic information from images. Since its advent, notable models like AlexNet (Krizhevsky, Sutskever & Hinton, 2012) and ResNet (He et al., 2016) have set new benchmarks in the field. Moreover, binary hash codes, known for their storage and computational efficiency, have become a key tool in addressing large-scale image retrieval challenges through the integration of deep learning with hashing techniques (Wang et al., 2018). The Convolutional Neural Network based Hashing rule (CNNH) (Xia et al., 2014) was one of the pioneers in introducing deep learning into the hash generation process, utilizing a two-stage method for generating hash codes, first by estimating hash codes through similarity matrix decomposition and then guiding model learning and hash code generation with these codes. The study by Zhao et al. (2015), Deep Semantic Ranking based Hashing (DSRH), moves away from using sample pairs to restrict the model suggesting direct integration of the convolutional network model with the hash function to leverage high-level semantic properties for mapping hash codes. Despite these advancements, most current deep hashing techniques (Li, Wang & Kang, 2016; Cao et al., 2017; Jiang & Li, 2018; Hoe et al., 2021) focus primarily on facilitating image retrieval for general categories such as cars or airplanes potentially overlooking the nuanced needs of fine-grained image retrieval in various practical scenarios. Therefore, the field of deep hashing (Cui et al., 2020; Jin et al., 2020; Ma et al., 2020; Wei et al., 2022) has recently begun to shift the focus on solving the challenge of fine-grained retrieval. This task targets precise retrieval of images from nuanced metacategories, emphasizing subtle distinctions among similar categories. Despite the remarkable achievements of these methods in fine-grained hash applications, the field still faces several challenges. (1) Components in different categories exhibit fundamental similarity, yet components in the same category exhibit wide diversity. This poses a challenge characterized by significant intra-class differences and minimal inter-class differences, thus limiting the validity of retrieval results. (2) Many models typically ignore the importance of features at different scales and in different dimensions when generating model hash codes and focus only on the final features. (3) The process of generating hash codes often leads to a reduced focus on important regions. Instead of effectively mapping these critical regions into the hash code, the process of generating the hash code generates interfering information and retains it in the hash code, and this interference tends to negatively affect the final results. In fact, the key to achieving fine-grained image retrieval lies in ignoring the problem of the dataset itself, acquiring richer and more discriminative high-level semantic features of the image and transforming them into hash codes with low loss, which can be used as a basis to guide the model to complete the retrieval.

This research aims to address the above challenges encountered in the field of fine-grained image retrieval. To this end, our research article introduces a novel deep neural network that utilizes a combination of feature fusion hierarchical cascading and self-attentive hashing. This model is crafted to effectively address the challenges of large-scale, detailed image retrieval tasks. The archietecture of our model in structured into two parimary segments: the offline feature compression phase and online feature ex-traction and hashing network. Within the feature extraction and hash coding phase, a hierarchical network structure based on ResNet50 is used. In order to amplify the significance of the extracted features, the ECANet mechanism is implemented on the most significant feature at each level of the cascade network. This mechanism encourages the model to prioritize the relationship weights of different channels on different feature dimensions. In addition, we combine these improved features with the original features extracted at each level to maintain critical feature information while avoiding the loss of auxiliary information. In the final phase of feature hash coding, the transformer architecture is applied following the combination of the extracted multilevel features, enhancing the hash code’s emphasis on the dataset’s overall features. This enable the model to more efficiently indentify important local regions within the optimized hash code. Furthermore, in order to enhance the extraced fine-grained features, we propose an attention-guided data im-provement approach, which specifically involves highlighting features at the center of the image and prompts the model to focus on areas exhibiting significant variances across different image categories. To further augment the model’s learning capabilities, we employ a multitask coordinated loss function for network optimization This method diverges from the conventional practice of manually setting weights in multitask learning by enabling the network to autonomously determine optimal weights. To affirm the superior performance and stability of our proposed model in detailed image retrieval, we have collected experimental results across various datasets. Initially, we executed comparative analyses against leading fine-grained image retrieval methods on five public and specialized datasets, employing significance testing to validate our model’s superior efficacy in these tasks. Subsequently, we extended our evaluation to five additional dataset types, demonstrating our model’s capacity to adeptly handle unseen data, with the retrieval accuracy solidly establishing its exceptional generalization capabilities. multidimensional ablation studies conducted on the CUB200, Aircraft, and NABirds datasets underscore the significant contribution of our model’s components to its overall performance. In summary, the primary contributions of this study are outlined as follows:

(1) To address the fundamental problem of fine-grained image retrieval, we propose an efficient solution on capturing image features of different dimensions and channels. Our strategy involves the development of a hierarchical network aimed at extracting image features from multiple dimensions. At each layer of our network, we design a two-branch feature summation module that combines attention-driven features with original features, thereby enriching the feature information while maintaining computational efficiency.

(2) In this study, we utilize self-attentive structure for the task of cascaded feature hash mapping. By adopting a distant feature guidance mechanism, our approach enables the model to learn features in the global focus region and facilitates the generation of efficient and compact hash codes. To optimize the use of limited available information, we also employ a multi-task learning loss scheme with weight self-learning to guide the training process.

(3) Detailed experimental analyses on five widely used fine-grained image datasets (CUB200-2011, Aircraft, Vegfru, Food101, and NABirds) demonstrate that our proposed method significantly outperforms other state-of-the-art methods, This achievement motivates researchers to further investigate into the benefits and properties of feature fusion within fine-grained image retrieval and the role of self-attention in hashing algorithms. Moreover, the efficiency gains in fine-grained image retrieval highlighted by our approach indicate its potential applicability to other visual tasks, meriting additional investigation.

The subsequent sections of this article are organized as follows. ‘Related Work’ provides a comprehensive assessment of the existing literature related to fine-grained image retrieval. ‘Methods’ elaborates on our proposed technique. ‘Experiments and Results’ outlines the experimental results and analysis. ‘Discussion’ discusses the experimental results. Finally, ‘Conclusions’ presents conclusions drawn from our study and potential directions for future research.

Related Work

This section provides an overview of the existing literature on content-based image retrieval, fine-grained image recognition, and fine-grained hash.

Content-based image retrieval

The main goal of content-based image retrieval (CBIR) is to identify images within a large database that bear similarity to a given query image (Knutsen Wickstrøm et al., 2022) .

In the early phases, scholars investigated several feature descriptors that were manually constructed and depended on visual cues such as colour, texture, shape, and other pertinent attributes to effectively describe images (Dubey, 2022). The specific image processing techniques mentioned include hue, saturation, value (HSV), red, green, blue (RGB), Zernike moment, Wavelet transform (Antonini et al., 1992), co-occurrence matrices (Gotlieb & Kreyszig, 1990), and contours and edges (Mehtre, Kankanhalli & Lee, 1997). Ramya put out a hybrid solution (Chen et al., 2022a), that attains enhanced retrieval efficiency. Sub-sequently, Wang, Yang & Li, (2013) and Rajkumar & Sudhamani (2019) conducted comprehensive studies to validate the efficacy of this approach.

Over the past decade, there has been a notable transition in feature representation and the efficacy of deep learning in the context of content-based image retrieval, resulting in significant advancements (Sharif Razavian et al., 2014; Jing & Tian, 2021). In 2011, Krizhevsky & Hinton (2011) were the first to employ deep autoencoders for the purpose of mapping images into concise binary codes, with the intention of facilitating content-based image retrieval (Zhong et al., 2016; Alzu’bi, Amira & Ramzan, 2017; Shen et al., 2020; Wang et al., 2020), Subsequently, numerous researchers and scholars from various countries and regions conducted multiple ex-periments to validate the viability of this approach (Zhong et al., 2016; Alzu’bi, Amira & Ramzan, 2017; Shen et al., 2020; Wang et al., 2020).

Content-based image retrieval methods have made some progress, but most of them focus on global features, when faced with fine-grained image retrieval problems that need to focus on local features, the existing content-based image retrieval methods cannot achieve the desired retrieval effect (Qian, Yu & Yang, 2023; Zeng et al., 2024).

Fine-grained image recognition

Current fine-grained recognition methods can be generally divided into three distinct classes (Wei et al., 2022): methods in-volving external sources of information, construction of end-to-end feature coding and integration of localised categorisation sub-networks.

Methods involving external sources of information has demonstrated effectiveness in many specific tasks, including web data analysis (Xie et al., 2015b; Xu et al., 2015; Cai et al., 2023) and multi-modal data processing (Xu et al., 2018; Song et al., 2020). The study referenced in Wang et al. (2020) is an early endeavour in utilising web data for training a mode (Niu, Veeraraghavan & Sabharwal, 2018). Researchers implemented a text encoder to enhance the accuracy of fine-grained image recognition, due to the high cost of labeling a large number of fine-grained image categories, this approach requires more auxiliary resources to train the model, thus leading to inefficient resource allocation. this approach required a greater amount of computing resource to train the model, hence resulting in an inefficient allocation of resources.

Local convolutional descriptors from the works of Xu, Yang & Hauptmann (2015), Cimpoi, Maji & Vedaldi (2015) and Gao et al. (2015) is an typical method of end-to-end feature encoding. In their study, Sun et al. (2018) developed a specialised loss function that facilitated the learning of multiple part-corresponding attention areas. This loss function aimed to bring same-attention same-class data closer together while simul-taneously pushing away different-attention or different-class features. The performance of the base network can generally be enhanced in comparison. However, these methods primarily emphasise global features, which may not be sufficiently effective for part-level (local) feature vectors and critical components.

The proficiency of the localization-classification subnetworks plays a crucial role in the process of feature extraction. Scholars developed models that effectively represent the distinguishing semantic components of things with fine-grained characteristics. Previous studies Girshick et al. (2014), Long, Shelhamer & Darrell (2015) and Ren et al. (2015) have utilised detection or segmentation algorithms in order to identify important sections within images. Researchers in studies Simon & Rodner (2015), Zhang et al. (2016) made an effort to utilise filter outputs as components for detecting parts. In their study, researchers Peng, He & Zhao (2018) and He, Peng & Zhao (2019) utilised the concept of attention to get hierarchical attention information. However, when this method is paired with other ways, its use cases are reduced, however it is an area that merits further investigation.

While the above approaches have yielded results in the field of fine-grained image recognition, inspired by the work of Sun et al. (2023) and Zhu et al. (2023) and others we believe that feature extraction efforts for the fine-grained image domain must take into account the impact of deep features from different scales and dimensions on the final result of the fine-grained task. The concepts we currently have are drawn from feature pyramids, small target detection and other academic studies, and in order to address the previously outlined problems we employ a top-down approach that incorporates features obtained from multi-scale dilation convolution, an approach that allows for the integration of complementary contextual information. In addition, we employ an attention-guiding mechanism following the introduction of cascading techniques to mitigate the problem of critical regions being overlooked.

Fine-grained hash

Fine-grained hash is an active research direction in recent years. Li, Wang & Kang (2016) addressed fine-grained hash problem and proposed a deep hash model, named deep pairwise-supervised hashing (DPSH), which performed simultaneous feature learning and hash code learning for applications with pairwise labels. Cao et al. (2017) presented a novel deep architecture called HashNet for deep learning to hash by continuation method with convergence guarantees, which learns exact binary hash codes from imbalanced similarity data. In 2020 and 2022, Xiu-Shen Wei team proposed ExchNet (Cui et al., 2020) and SEMICON (Shen et al., 2022) worked on the local features to speed up fine-grained hash, reduce storage and promote availablility respectively. Zeng & Zheng (2023) designed FGC-Net by modifying the network structure of FISH (Chen et al., 2022b), and their experiments showed that the cascade network can effectively improve the retrieval efficiency. Since 2023, the target localization module was introduced into retrieval networks, the main object was separated from the background by selecting convolutional feature descriptors, which effectively improved the performance of fine-grained image retrieval by filtering out most of the interfering information (Wang, Zou & Wang, 2023). The following proposed a fine-grained image retrieval method (AMCICIR) based on the attentionl mechanism and the constraints of contextual information, firstly, gradually refining the target localization through the attention learning mechanism, and extracting useful local features from coarse to fine. AMCICIR acquires valuable local features in a progressive manner by utilizing an attention learning mechanism to refine the target localization from coarse to fine. Afterwards, an improved graph convolutional network (GCN) is used to model the internal semantic interactions of the learned local features to obtain a more discriminative and fine-grained image representation (Li & Ma, 2023). Later others replaced attention-guided features with convolutional descriptors and proposed an Attribute Searching and Mining Hash (AGMH), which groups category specific visual attributes and embeds them into multiple descriptors to generate a comprehensive feature representation for efficient fine-grained image retrieval (Lu et al., 2023). Later Duan et al. (2022) computed similarity matrices at channel, pixel, and spatial levels by acquiring deep localized descriptors. Based on the computed comprehensive similarity matrix, a multilevel similarity-aware loss function is proposed using the deviation between pairwise distances and violated margins, which made full use of the information samples and thus achieved a better matching of the true similarity relationships between classes.

The above methods have made significant contributions to the field of fine-grained image retrieval. However, their main goal is still focusing on the localization of the target’s key regions and incorrectly associated features with the number of valid bits of the abbreviated hash code, which lead to the limitation of the retrieval efficiency and becamde a key issue that needs to be paid attention to (Lu et al., 2023). This study draws inspiration from the Transformer model architecture to effectively capture the complex details of the overall context. By using the self-attention module, we investigate the impact of various feature regions on the effective mapping of hash codes before generating the final hash code. Thus, this research provides new insights to address the above problem.

Methods

This research primarily focuses on the integration of fine-grained hash learning into convolutional neural network (CNN) models. Additionally, a novel neural network architecture has been presented in order to direct the model’s attention towards characteristics at various scales and exploit interrelationships for the purpose of generating enhanced hash codes. As depicted in Fig. 1 the methodology is bifurcated into two main phases: an offline feature compression phase and an online feature hash learning phase utilizing neural networks. In the first stage, the model converts the input raw image data into an accurate binary hash code that conforms the neural network model’s learned format and the predefined length of the input image. The outcomes of these processes serve as the benchmark for the model’s subsequent learning phase. In the second stage, the deep learning method of multiple forward and backward passes is used to minimize the discrepancy between the target value and the analyzed value of the model as much as possible, and the optimal parameters of each stage of the model are saved and used as the final structure.

Figure 1 Our proposed contains two steps: an offline feature compression phase and an online feature hash learning phase utilising neural network.

Image credit: the CUB200-2011 archive at https://www.vision.caltech.edu/datasets/cub_200_2011/.

Offline feature compression

Traditional image retrieval methods primarily adopt an instance-to-instance approach for metric learning, with prevalent techniques including pairwise and triplet loss. These methods can significantly increase the computational load of the model as they enhance its data mining capabilities, relying heavily on various optimization strategies. Movshovitz-Attias et al. (2017) and Kim et al. (2020) demonstrated that agent-based methods can substantially enhance convergence speed and robustness without sacrificing precision. Later (Zeng & Zheng, 2023) described a similarity relation method between mapped instances and the agent vector of each class for fine-grained hash learning, which yielded promising results; therefore, this research employs an enhanced approach based on mapping proxy vectors. This section describes the mapping procedure from training data to hash codes. It is noted in Fine-graIned haSHing (FISH) that the task of obtaining a reasonable binary proxy code is to optimize the following equations: (1) minM,N||Y−NM||F2,M∈−1,1k×n.

The following equation can be obtained for the purpose of optimization by adding an intermediate state of and an orthonormal rotation matrix that makes and as similar as possible, where is the dynamic parameter: (2) minM,N,O,P||Y−NO||F2+λ||M−PO||F2,M∈−1,1k×n,PPT=I.

Where, in optimizing the parameter M in Eqs. (3)–(2), assuming that the other variables remain constant, Eq. (2) can be redefined as: (3) minM,N,O,PTrM−POT×M−PO,M∈−1,1k×n.

Since TrMTM isa specified constant, Eq. (3) can be optimised as follows: (4) minM−TrMTPO,M∈−1,1k×n.

By optimising parameter Nwhile holding the other variables constant, Eq. (2) can be transformed into: (5) N=YOTOOT−1.

Equation (2) is equivalent when optimising parameter O with the other variables held constant. (6) O=NTN+λPTP−1×NTY+λPTM.

Solving the problem of optimizing P with all other variables held constant can be accomplished by singular value decomposition (SVD), or demonstrating that (7) MOT=SΩS ~T⇒P=SS ~T.

It can now be demonstrated that Eq. (1) can be optimised, i.e., the class’s binary agent coding agent mapping can be implemented. Eventually, by iterating L times Eq. (1), the final optimised M matrix as the trained set of agent encoding can be obtained.

Online feature hash learning

This section delves into the pivotal function of the second phase of the proposed program, this research merges and enhances ECANet (Wang et al., 2020) and ResNet50-based cascade network, and embeds a self-attention mechanism in generating hash codes to form a Multi-FusNet structure in order to mine fine-grained image features more accurately and comprehensively, and to realize streamlined coding. The proposed network is seen in Fig. 2.

Figure 2 An overview of the proposed which uses the fused image feature approach and the multi-head attention with multi-task balanced loss method for hashing.

Image credit: the CUB200-2011 archive at https://www.vision.caltech.edu/datasets/cub_200_2011/.

Single-level feature fusion module

We use a common pre-trained ResNet50 as the foundational architecture of a fully convolutional neural network dedicated to extracting image features given ResNet’s proficiency in feature extraction within the realm of image classification. In addition to enhance the initial feature information coverage, we extracted the feature maps of different layers, which are represented as X = {x1, x2, x3, …, xj}.

In the context of deep feature extraction, it is evident that different channels have the capacity to capture diverse features. Specifically, in the analysis of fine-grained images, applying uniquely extracted weights to each channel within the feature layer ensures the feature map is dynamically influenced by these adaptive channel weights during extraction. This approach allows layers with more crucial features to exert a stronger influence on the outcome. In 2018, a study by Hu, Shen & Sun (2018), SENet (Squeeze-and-Excitation Networks), presented evidence that assigning different weights to various channels during image feature extraction could significantly improve the network’s image representation capabilities. Nonetheless, while this method does enhance image representation, it can also adversely affect model accuracy and computational efficiency due to the dimensionality reduction step involved. Addressing this, ECANet introduces a local cross-channel interaction method that eschews dimensionality reduction which obtains aggregated features through global average pooling, and reinforces the features by bootstrapping the channel weights from k. This strategy effectively mitigates the negative impact of dimensionality reduction on channel attention during the learning process while preserving the advantages of SENet.

As the depth of the network increases, some critical features tend to diminish. in order to preserve the both the most and second most important information for an extended duration, we employ a two-branch approach that maintains the original features and uses a channel-guided feature-based method, respectively. This strategy allows for the extraction of more significant information without expanding dimensionality through straightforward pixel stacking. This process is articulated as follows: (8) Y=x1+ecax1,x2+ecax2,⋅⋅⋅,xj+ecaxj.

Following the completion of each level of feature fusion, a cascade operation is executed on all levels of extracted features in order to produce the ultimate features that facilitate the hash mapping. The specific arithmetic formulas for this process are as follows, and in addition, the specific combinations in this section can be found in Fig. 3. (9) Y ˆ=concaty1,y2,y3,…,yj.

Figure 3 Our proposed dual-channel fusion strategy of ECANet enhanced and original feature fusion in a single layer of the feature cascade.

Hash-feature enhancement module

In our proposed approach for fine-grained hash job, we leverage the capabilities of Multi-headed self-attention. The structural representation of this technique is illustrated in Fig. 4, which facilitates the improvement of hash code generation. In ‘Single-level feature fusion module’, a collection of fused intermediate features, denoted as Y=y1,y2,y3…,yj, , is obtained. These features are then stitched together into a one-dimensional vector, referred to as Y ˆ∈iH×W×1. This vector is subsequently fed into the hash feature enhancement module for the purpose of further optimising the features. The utilisation of the multi-headed attention mechanism in image feature mapping has greatly contributed to various aspects such as capturing global context, learning feature interactions, ensuring robustness to changes, and merging multi-scale information. Additionally, the transformer structure, referred to as trans(⋅), serves as an enhancement module by effectively extracting relevant features that can realise long-range feature association without significantly increasing the model complexity. By employing a multi-part approach, the model is capable of selectively focusing on different feature layers. This enables the model to effectively capture the overall structure and relationships among visual components. Additionally, the model retains the ability to preserve fine-grained information by considering both local details and global context. This is crucial for generating hash codes that are both accurate and concise.

Figure 4 Multi-head structure contains two components: (A) scaled dot-product attention and (B) multi-head attention.

The final hash mapping module can be described as a two-layer linear mapping with the inclusion of the Exponential Linear Unit (ELU) activation function. This module generates the feature vector for the hash code. During the training phase, the code replaces 0 with -1 to facilitate mathematical operations. The full procedure is depicted as follows: (10) code=LELUBLtransY ˆ.

L(⋅) represents the ordinary linear mapping function, B⋅ represents the normalization layer, and ELU(⋅) denotes the activation function, and the formula is as follows: (11) fx=x,x≥0αex−1,x<0.

Multi-task balanced loss

Designing effective loss functions to guide the learning process is a crucial aspect of fine-grained hash. The combination of categorical and hash losses balances the loss function to address the inherent difficulties of fine-grained datasets, such as unbalanced data distributions and varying importance levels between fine-grained characteristics, thereby ensuring the fair and efficient optimization of fine-grained hash models.

In particular, the combination of classification loss and hash loss can optimize classification precision and hash code generation. The classification loss encourages the model to acquire discriminative representations that accurately classify fine-grained categories, while the hashing loss promotes the generation of compact binary codes that preserve semantic similarity between visually similar categories. By integrating the two into a balanced loss function, the model can be trained to capture discriminative detals enabling the generation of hash codes that accurately represent fine-grained visual content, thereby enhancing precise retrieval capabilities. Nevertheless, it is frequently the case that multitasking loss functions necessitate manual adjustment of the contribution ratio of different branches to the overall loss in various application contexts, significantly diminishing the effectiveness in addressing the related issues. Informed by the literature (Shen et al., 2022), we redesigned a loss function with adaptive multi-task weights, the formula for which is shown below, where α and β represent the learnable hyperparameters, respectively. (12) Lres=1αLhash+1βLcls+ logα+1+ logβ+1.

A unified proxy for a class is more likely to capture class-level features and ignore noise than a single instance, as a result of which we design a loss function Lhash that is better suited to the task of fine-grained hash, which coincidentally coincides with the approach proposed by FISH. The basic idea of agent-based loss is to maintain the similarity between an agent and its instances within a corresponding subset, where the agent is specifically a representative of a subset of the training data and participates in learning as part of the network parameters. For example, sim⋅ isused to measure the similarity between two vectors, p1,p2,…,plare the agent vectors of class l, and x is the data vector belonging to class i. Thus, the agent-based loss can be expressed as: (13) Lproxy=−logexpsimx,pi∑j≠i expsimx,pi.

The formula attempts to keep xas close as possible to the plwhile differentiating it from other agent vectors. Such agent-based loss functions are primarily used for retrieval tasks with real-valued features but present optimization challenges in hashing tasks due to binary constraints. To make the hashing task feasible, we adpot a straighforward yet effective loss function: (14) Lhash= ∑j=1l ∑i=1nYji−pjbi2.

where yij represents the learning matrix, bi is the hash code of a single instance, and p ∈ il×k is the agent vector of the j-th class, which has not been restricted to binary form for the purposes of optimisation.

For the classification task, we used the standard cross-entropy as the loss function. To produce a better calibrated network, we introduced label smoothing transitions with a confidence level of 0.90 and used both the original and Gaussian-enhanced images as optimisation perspectives. The formulation is given as follows which yi∈0,1: (15) Lcls=−∑i=1n0.1×yi+0.9l× log expy¯i ∑k=1l expy¯ik2.

In order to solve the problem that the hash code binary constraints cannot be optimised directly with the back propagation algorithm in Eq. (12), we combine all the agent vectors into a matrix DϵRl×k and bring the hash code matrix C=bii=1nϵ−1,1k×n to be optimised into Eq. (14), and the optimisation yields the following formula,where ∥⋅∥F denotes the Frobenius norm of the matrix. (16) Lhash∗= ∑j=1l ∑i=1nY−DC2+ ∑i=1n||bi−WbCi⊙Rfi||F2.

Eventually we transformed Eq. (14), into a two-part equation showed in Eq. (16), where the left half solves the optimisation problem for learning hash codes, and the right half can be solved by a conventional backpropagation algorithm similar to Eq. (16). Consequently, the overall optimization of hash loss is segmented into two phases. The first step is the optimisation of the left half of Eq. (16) before network training executed through alternating iterations before training the network as described in ‘Offline feature compression’. The second step is to optimise the hash problem together with the classification problem during network training, where the hash loss function is the right half of Eq. (16).

Experiments and Results

In this section, we undertake a thorough series of rigorous tests to evaluate the performance of our research in comparison to various modern hashing algorithms. The aforementioned tests are performed on a variety of benchmark datasets. The following analysis aims to assess the distinct contributions of the individual modules introduced. Moreover, we will explore how the integration of various information modules at the pinnacle of each cascade influences the overall outcomes. Additionally, visualisation techniques will be employed to evaluate the effectiveness of the proposed feature fusion procedures. We will also examine the network’s convergence across multiple datasets. In sum, we will showcase the effectiveness of our proposed method through a real-world prediction scenario.

Datasets and evaluation metric

We evaluate the proposed approach on several fine-grained datasets listed in Table 1: (1) CUB-200-2011 (Wah et al., 2011) contains 11,788 images of 200 bird species, and the dataset is divided into: training set (5,994 images) and test set (5,794 images), (2) FGVC Aircraft (Maji et al., 2013) contains 10,000 images of 100 aircraft models, the dataset is divided into: training set (6,667 images) and test set (3,333 images), (3) VegFru (Hou, Feng & Wang, 2017) contains 160,731 images, covering 200 vegetable categories and 92 fruit categories, the dataset is divided into: training set (29,200 images) and test set (116,931 images), (4) Food101 (Bossard, Guillaumin & Van Gool, 2014) contains 101 kinds of food , 101,000 images, the dataset is divided into: training set (75,750 images) and test set (25,250 images), (5) NABirds (Van Horn et al., 2015)) contains 555 kinds of birds, 48,562 North American image datasets divided into training set (24,633 images) and test set (23,929 images).

Table 1 Five fine-grained image benchmark datasets.

Datasets	Comparison	
	CUB	Aircraft	Food101	NABirds	VegFru	
Database	11,788	10,000	101,000	48,562	160,731	
Training	5,994	6,667	75,750	24,633	29,200	
Testing	5,794	3,333	25,250	23,929	116,931	
Species	200	100	101	555	290	

In order to further demonstrate the applicability of our proposed method we performed method validation with the five datasets listed in Table 2. The five datasets are:

Table 2 Five generalization capability validation datasets.

Datasets	Generalization	
	Standdogs	Flowers	Animals	Buildings	BJFU100	
Database	20,580	4,322	5,400	5,063	10,000	
Training	12,000	3,459	4,320	4,057	8,000	
Testing	8,580	863	1,080	1006	2,000	
Species	120	5	90	17	100	

(1) Stanford dogs (Khosla et al., 2011), which contains images of 120 breeds of dogs from all over the world, of which 12,000 are included in the training set and 8,580 are included in the test set.

(2) Flowers (Nilsback & Zisserman, 2008), which contains 4,322 photos of flowers mainly collected from flicr, google images, yandex images, of which 3,459 are in the training set and 863 are in the test set.

(3) Banerjee (2022), which contains 90 different categories of animals, with about 5,400 animal images, each category contains 60 images, of which 4,320 are in the training set and 1,080 are in the test set.

(4) Oxford 5k Building (Philbin et al., 2007), which consists of about 5,000 images of buildings of 17 species, and the dataset is used by a large number of retrieval systems to measure the quality of a building. The dataset is used by a large number of retrieval systems to measure the system performance, of which the training set is 4,057 and the test set is 1,006.

(5) BJFU100 (Sun et al., 2017), the dataset is collected from natural scenes by mobile devices, including 100 species of ornamental plants on the campus of the Beijing Forestry University (BFU), and each category contains one hundred different photographs, of which the training set is split into 8,000 pictures, and the test set is 2,000 pictures.

The mean average precision (mAP) evaluation index is employed for the purpose of assessing the retrieval performance. Where Q represents the total number of queries instances, APi is the Average Precision for each query instance, P@k is the precision at the given rank k in the ranked list of retrieved items. rel@k is an indicator function that represents whether the item at rank k is relevant or not. It takes a value of 1 if the item is relevant and 0 otherwise. G is the total number of relevant items for the query instance. K is the total number of retrieved items in the ranked list, the detailed formula is shown below: (17) mAP=1Q∑i=1QAPi.

Implementation details and experimental settings

Experimental settings: Our implementation of Multi-FusNet is based on the Torch library and the Pytorch framework. To ensure a comprehensive evaluation in line with existing deep models in the field, we employ a fundamental model that aligns with the comparison experiments as a baseline to execute and authenticate our approach. The tests employ a pre-trained Resnet50 as the underlying framework. During the experimental testing, it is sufficient to solely consider the category label content of the images for the purposes of this study. No additional information, such as border annotations, is required.

Implementation details: Regarding data preprocessing, a set of standardised image data augmentation procedures were implemented in order to address the disparities among various datasets. This process is illustrated on the left side of Fig. 5. During the training procedure, the input image undergoes a random cropping operation to achieve a size of 336  × 336. Subsequently, a random rotation within a range of 30° is applied to the image. The process also involves generating an extended output of the original image. Finally, a random horizontal flip is performed with a probability of 0.7, accompanied by adjustments to the brightness, contrast, saturation, and hue of the original image with a factor of 0.5. During the testing phase, the dataset was uniformly shrunk to an image size of 356  × 356. Subsequently, the images were centred and changed to a size of 336  × 336 in order to serve as input for the model. A stochastic gradient descent (SGD) optimizer is employed, with a momentum value set to 0.91. The weight decay is configured with a value of 0.0005, while the number of iterations is specified as 300. In all datasets, a uniform starting learning rate of 0.0035 was employed, along with a mini-batch size of 128. Regarding the offline extraction of the class hash code part of our main set of 15 iterations of compression, the rest of the specific method can be referred to ‘Offline feature compression’ of this article, the specific process can be seen in Fig. 5, the right side of the pseudo-code section.

Figure 5 Basic several image preprocessing operations with offline feature compression process pseudo-code.

Image credit: the CUB200-2011 archive at https://www.vision.caltech.edu/datasets/cub_200_2011/.

Main results

Comparison with the SOTA methods: In our experimental setup, we utilised seven well-established approaches (namely DPSH (Li, Wang & Kang, 2016), HashNet (Cao et al., 2017), ADSH (Jiang & Li, 2018), ExchNet (Cui et al., 2020), AA-Net (Chen et al., 2018), SEMICON (Shen et al., 2022) and AMGH (Lu et al., 2023)) as benchmarks to showcase the effectiveness of our suggested approach. In this part, Table 3 displays the mean average precision (mAP) for the five publicly available datasets discussed in ‘Datasets and evaluation metric’ section. The evaluation is conducted using various hash bit dimensions (12, 24, 32, 48). The greatest MAP value is indicated in bold.

Table 3 mAP results for different number of bits on five fine-grained datasets.

Datasets	#bits	SDH	DP SH	Hash-Net	AD SH	Exch-Net	AA-Net	Semicon	AM GH	Ours	
CUB200-2011	12	10.52	8.68	12.03	20.03	25.14	33.83	37.76	56.42	72.02	
24	16.95	12.51	17.77	50.33	58.98	61.01	65.41	77.44	78.70	
32	20.43	12.74	19.93	61.68	67.74	71.61	72.61	81.95	82.67	
48	22.23	15.58	22.13	65.43	71.05	77.33	79.67	83.69	84.69	
Aircraft	12	4.89	8.74	14.91	15.54	33.27	42.72	49.87	71.64	74.74	
24	6.36	10.87	17.75	23.09	45.83	63.66	75.08	83.45	85.27	
32	6.90	13.54	19.42	30.37	51.83	72.51	80.45	83.60	87.35	
48	7.65	13.94	20.32	50.65	59.05	81.37	84.23	84.91	88.20	
Food101	12	10.21	11.82	24.42	35.64	45.63	46.44	50.00	62.59	79.04	
24	11.44	13.05	34.48	40.93	55.48	66.87	76.57	80.94	81.43	
32	13.36	16.41	35.90	42.89	56.39	74.27	80.19	82.31	82.55	
48	15.55	20.06	39.65	48.81	64.19	82.13	82.44	83.21	84.03	
NABirds	12	3.10	2.17	23.40	2.53	5.22	8.20	8.12	——-	73.19	
24	6.72	4.08	32.90	8.23	15.69	19.15	19.44	——-	79.96	
32	8.86	3.61	45.20	14.71	21.94	24.41	28.26	——-	81.10	
48	10.38	3.20	49.70	25.34	34.81	35.64	41.15	——-	82.11	
VegFru	12	5.92	6.33	3.70	8.24	23.55	25.52	30.32	43.99	79.94	
24	11.55	9.05	6.24	24.90	35.93	44.73	58.45	68.05	81.03	
32	14.55	10.28	7.83	36.53	48.27	52.75	69.92	76.73	84.54	
48	16.45	9.11	10.29	55.15	69.30	69.77	79.77	84.49	85.79	

Based on the aforementioned comparison results, it is evident that our proposed approach has substantial effectiveness across all five fine-grained datasets, and our approach demonstrates superior performance compared to current widely-used techniques for mapping short hash codes of 12 bits and 24 bits. Specifically, our experiments on the NABirds dataset indicate a significant improvement in performance by a factor of 9. We attribute this improvement to the model’s exceptional ability to extract multi-level feature information.

The performance of our approach in mapping longer hash codes (32 bits and 48 bits) compared to shorter hash codes is relatively small and more influenced by the dataset. For example, in the Food101 dataset, the improvement using a 48-bit hash code was only 2 percent. However, in the NABirds dataset, the experimental improvement using 48-bit hash codes is as high as 40%. We attribute this difference mainly to differences in the characteristics of the datasets used. Food101 has more categories and more samples per category than the NABirds dataset, but the training images all contain some degree of noise. NABirds, on the other hand, has fewer categories and fewer samples per category, but each category has an associated independent annotation. In addition, we used image enhancement techniques based on Gaussian curve magnification in addition to the original image for regression optimisation in the loss function design, and observed that the subtle differences between the targets in this dataset of NABirds were mainly concentrated in the centre of the image, whereas those in Food101 were clearly spread across the whole image. We therefore conclude that the image enhancement technique we employ has superior retrieval capabilities for datasets where the differences are distributed in the centre. This conjecture is also confirmed by the VegFru dataset, where we find that the targets are also located in the centre of the image, and the retrieval results are also significantly improved compared to the other methods. The experimental results demonstrate that our proposed approach consistently achieves a retrieval accuracy of over 70%. This holds true across various scenarios, including both limited and large-scale fine-grained datasets. Additionally, the approach performs well with both short and long hash code mappings. Notably, the long hash mapping of 48 bits in the Aircraft dataset achieves an impressive 88% of the mAP value. These findings provide strong evidence supporting the feasibility and effectiveness of our approach. Furthermore, the outcomes derived from the various datasets examined demonstrate that our model attains an average enhancement of 40% at 12-bit, 22% at 24-bit, 16% at 32-bit, and 11% at 48-bit in comparison to the prevailing SEMICON methodology. Compared to the state-of-the-art AMGH method, our model achieves an average improvement of 50%, 27%, 21% and 14% for 12-bit, 24-bit, 32-bit and 48-bit, respectively, the aforementioned findings provide compelling evidence on the efficacy of our approach.

In order to avoid that the superiority of our proposed method over the current state-of-the-art schemes occurs by chance, we use a statistical significance test in the form of a paired-sample t-test to statistically analyze our proposed method against the two current state-of-the-art methods of fine-grained image retrieval for all the datasets listed in Table 1, and the results of the test are shown in Table 4.

Table 4 Results of paired-sample t-test analyses with the two most recent mainstream fine-grained image retrieval methods.

		Pairing difference				
		Mean	Standard deviation	Standard error of mean	95% confidence interval for difference	t	df	sig	
					Lower-bound	Upper-bound				
Comparison 1	Semicon–Ours	−22.932	20.785	4.648	−32.66	−13.204	−4.934	19	0.000	
Comparison 2	AMGH–Ours	−6.661	9.555	2.389	−11.753	−1.560	−2.789	15	0.014	

As can be seen in Table 4, the two-tailed test P-value of Semicon and our proposed model structure is 0.000 < 0.01, which is a significant difference occurring at the confidence level of less than 0.01. In addition, the two-tailed test P-value of the newly proposed AMGH method in 2023 and our proposed method is 0.014 < 0.05, which indicates that the two methods are occurring at the level of less than 0.05 Significant difference between the two methods occurs at a level less than 0.05. In summary, it can be concluded that our proposed method has a more significant advantage over other methods in fine-grained image retrieval tasks and this advantage does not happen by chance.

Generalization experiment: In this section, in order to evaluate the performance of the model more comprehensively and objectively, and to ensure that the model has good adaptability to data from different sources and under different conditions, we do the retrieval experiments of 12 bits, 24 bits, 32 bits, and 48 bits in the five sets of datasets listed in Table 2, respectively. In Table 5, we directly record the actual mAP retrieval results of the model for different hash lengths in different datasets, and in Fig. 6, we use bar charts and line graphs to visualize the differences between different categories of the model and the trend of the results.

Table 5 mAP results of our proposed method on five datasets.

	12 bits	24 bits	32 bits	48 bits	
Standdogs	77.42	78.73	79.65	80.19	
BJFU100	96.98	98.06	98.49	98.61	
Flowers	95.51	95.88	96.48	96.73	
Animals	90.01	91.60	92.55	93.86	
Buildings	65.31	65.95	67.09	68.65	

Figure 6 Results of model generalization experiments on five datasets with code lengths ranging from 12 to 48.

As listed in Table 5, the experimental results in general range from 65.31 (Buildings, 12 bits) to 98.61 (BJFU100, 48 bits), but the overall ability to meet the basic requirements of the retrieval task indicates that our proposed model has excellent generalization ability in both fine-grained (Standford dogs, Oxford Buildings) and coarse-grained BJFU100, Flowers, Animals datasets, and both multi-species (Standford dogs, Flowers, Animals) and less-species (Flowers) datasets. BJFU100, Flowers, Animals datasets, and both multi-species (Standford dogs) and few-species (Flowers) datasets have excellent generalization ability. However, Fig. 6 also clearly demonstrates the significant differences in retrieval results between different hash lengths under different datasets, for example, as shown in the bar chart, under all digits, BJFU100 has the highest results, while Buildings has the lowest results, and although the results of the other three datasets are in the middle of the pack but also show a stable order overall. In addition as shown in the line graph, all datasets have improved results when the number of hash bits increases this finding is consistent with the assumption of a fine-grained image retrieval task. However, the gap between the results of different datasets with the same hash code length is more prominent, for example, the retrieval results of BJFU100, Flowers, and Animals are in the upper part of the line graph, while Standford dogs is always in the middle of the line, and Oxford Buildings retrieval results are much lower than the other datasets are more obvious. By further investigating the differences between several datasets, we find that the differences between different categories in the BJFU100, Flowers, and Animals datasets with excellent performance are more obvious, and it is favorable for the network model to obtain features in terms of the overall image features, the pose of the target, and the complexity of the background of the image, although the three datasets listed do not belong to the fine-grained datasets, but they are excellent in retrieval. The results also verify the outstanding generalization ability of our proposed model in the image retrieval domain. The Stanford dogs and Oxford Buildings datasets are limited by the shooting angle, light and background complexity, and there is the problem of large intra-class differences in fine-grained images and small inter-class differences, and the overall worse results than the other three datasets are within an acceptable range. It was found that the significant differences between the two and the poorer results compared to the other fine-grained image datasets were also due to the presence of a large number of images in the dataset with small targets, as well as negative data unrelated to category. This finding is illustrated in detail in Fig. 7, for example, images of people holding dogs, people in front of buildings and even other image data that contain no category association. Overall our proposed model has excellent generalization ability in the field of fine-grained image retrieval especially image retrieval but at the same time this ability is severely limited by the quality of the images in the dataset.

Figure 7 Sample of datasets, where one row of three images in each dataset is a category, and the red boxes show the images that negatively affect retrieval accuracy.

Image credit: the Stanford dogs archive at http://vision.stanford.edu/aditya86/ImageNetDogs/main.html, the Oxford Building archive at https://www.robots.ox.ac.uk/ vgg/data/oxbuildings.

Ablation studies

Effectiveness of modules:In order to verify the validity of the proposed model components, we decompose Multi-FusNet and conduct ablation experiments accordingly. The model consists of three key components: the hierarchical network module “Cascading”, the feature fusion module “Fusion” and the hash code mapping module “Attention”. The efficacy of our proposed three main modules is demonstrated through ablation experiments on three publicly available fine-grained image datasets, including CUB200-2011, Aircraft, and Food101. In the ablation experiments, we sequentially incorporate the three modules into the ResNet50 backbone network as the experimental baseline. The evaluation of our proposed network Multi-FusNet is based on the mAP metric mentioned in ‘Datasets and evaluation metric’. The results consistently show that our method improves the mAP metric for different datasets and hash bit lengths through module stacking, and the results can be seen in Table 6 and Fig. 8.

Table 6 Retrieval accuracy (%mAP) with incremental modules of the proposed model.

Configu-Rations	CUB200-2011	Aircraft	NABirds	
Resnet50	Cascading	Fusion	Attention	12	24	32	48	12	24	32	48	12	24	32	48	
✓	X	X	X	51.33	54.48	56.89	59.88	50.43	55.94	58.42	60.66	51.74	54.83	58.99	60.01	
✓	✓	X	X	70.10	75.43	78.09	83.92	70.09	82.13	80.15	82.17	70.41	77.36	78.95	80.42	
✓	✓	✓	X	71.92	77.04	79.10	84.10	73.90	83.14	85.11	86.4	72.14	78.10	79.93	81.79	
✓	✓	X	✓	70.88	77.10	78.11	83.90	72.71	83.15	84.98	85.17	72.10	78.04	79.88	81.38	
✓	✓	✓	✓	72.02	78.70	79.97	84.69	74.74	85.27	87.35	88.20	73.19	79.96	81.10	82.11	

Figure 8 The mAP results of our proposed method with each method with different hash lengths for different datasets.

In order to avoid the experimental results are caused by chance, the study uses paired-sample t-tests to determine the effect of the modeling method we used and the method after deleting modules one by one on the results of the three datasets of CUB, Aircraft, and NABirds under each hash length, and the results show that the overall retrieval accuracy of the model using our proposed method and after deleting any module shows a difference in 0.01 significance at the 0.01 significance level. The results show that the overall retrieval accuracy of the model using our proposed method and after deleting any module is different at the 0.01 significance level, and further comparison of their overall t-test results reveals that our proposed model is higher than several other structures, which fully proves the validity of each of our proposed structures, as shown in Table 7.

Table 7 Pre-test-post-test paired sample t-tests.

		Pairing difference					
		Mean	Standard deviation	Standard error of mean	95% confidence interval for difference	t	df	Sig	
					Lower-bound	Upper-bound			One-sided	Two-sided	
Comparison 1	Line1 - Line5	−24.475	2.862	0.826	−26.293	−22.657	−29.624	11	<.001	<.001	
Comparison 2	Line2 - Line5	−3.173	1.891	0.546	−4.375	−1.972	−5.815	11	<.001	<.001	
Comparison 3	Line3 - Line5	−1.219	0.711	0.205	−1.671	−0.767	−5.936	11	<.001	<.001	
Comparison 4	Line4 - Line5	−1.658	0.691	0.199	−2.097	−1.219	−8.314	11	<.001	<.001	

Figure 8 shows that our method outperforms the previously proposed fine-grained image retrieval methods in the three datasets listed in terms of mAP values for each hash length, especially in the field of short hash codes, which is significantly better than the historical methods, and we believe that this is related to our proposed method’s ability to mine and integrate the features at different scales of the image, and this conjecture is also proved in the NABirds dataset in figure. In this dataset, only the FGCNet method retrieval results are similar to ours. This similarity arises from its utilization of a cascaded residual network. However, it diverges from our method by ignoring the deep information such as image features at each level and the long range concerns of hash code mapping after feature integration. In summary, it can be directly proved that our method outperforms the current retrieval model by Fig. 8. As can be seen from Table 6, the two-tailed test P-value of Base experiment vs. our proposed method, adding residual structure vs. our proposed method, adding feature fusion vs. our proposed method, and adding long-distance hash code mapping module vs. our proposed method are all less than 0.001, which indicates that the significant difference occurs at the level of less than 0.001. In summary, the contribution of each key module of our proposed structural model to the overall effect improvement is very significant.

Effect of different attention: In this section, we conduct comparative experiments with many commonly used attention modules to illustrate the superior performance of efficient channel characteristics in mining critical information across multiple layers for enhancing retrieval efficiency. In addition to the conventional channel attention module SENet, as discussed in ‘Single-level feature fusion module’, our approach incorporates several other channel attention modules. These include the CANet (Hou, Zhou & Feng, 2021) module, which incorporates location information, the CBAM(Woo et al., 2018) module, which integrates spatial and channel attention, and the PSA (Liu et al., 2021) module, which employs feature pyramid grouping to extract features at various scales. The experiments are conducted as follows: the ECA module in the original scheme is replaced with other fusion strategy attention approaches, while maintaining the same hyperparameter configuration. The performance of each combination scheme is analysed by evaluating the mAP values. The studies were performed on three publicly accessible fine-grained image datasets, specifically CUB200-2011, Aircraft, and NABirds. The performance of hash code mapping was evaluated using 12-bits, 24-bits, 32-bits, and 48-bit representations, and a comparative analysis was conducted. To facilitate the visualisation of the experimental outcomes, histograms depicting the mAP values are constructed for each method across various hash lengths and datasets. The specific results are presented in Fig. 9. In this figure, the horizontal axis represents different bit lengths within a given dataset, while the vertical axis represents different datasets for a specific bit length. Additionally, each graph consists of five columns, with the first column representing SENet, the second column representing CBAM, the third column representing CANet, the fourth column representing prostate specific antigen (PSA), and the last column representing SENet. The first column denotes the PSA, whereas the final column signifies the utilisation of the efficient attention module (ECA).

Figure 9 Influence of different attention on results.

The results demonstrate that the utilisation of our efficient channel-focused modules yields exceptional outcomes across all datasets and hash mapping lengths. Specifically, the aforementioned modules exhibit significantly superior performance compared to the other modules, particularly when dealing with small hash maps of 12 bits and 24 bits. Furthermore, it is observed that CBAM and PSA exhibit superior performance compared to SENet and CANet in practical applications. This trend is particularly evident when considering the Aircraft dataset, particularly for hash mappings of shorter length. Furthermore, it has been observed that CBAM and PSA algorithms can exhibit comparable performance to the efficient channel note in certain scenarios. For instance, in datasets such as CUB200-2011 (24 bits), Aircraft (12 bits), and NABirds (48 bits), these algorithms have demonstrated promising results. This finding suggests the potential for further exploration into the impact of features beyond channel information on fine-grained feature hashing. The aforementioned experiments effectively showcase the notable impact of employing efficient channel attention in the context of fine-grained image retrieval. Additionally, our findings suggest that further investigation into the utilisation of channel attention, in conjunction with spatial or multidimensional information, holds promise for enhancing retrieval performance.

Visualization feature fusion effect

The class activation map (CAM) (Zhou et al., 2016) is a visual representation that highlights the regions of images that are most influential in the decision-making process of a neural network. This allows for a deconstruction of the opaque decision-making process of the network, aiding in the comprehension of the model’s decision-making mechanisms. The feature heatmap for the two-branch feature fusion utilised in this study is presented in Fig. 10. The first column represents the original image, indicated by the red box. The second to fourth columns display the original feature visualisation, the ECA-enhanced feature visualisation, and the two-branch fusion feature visualisation, respectively.

Figure 10 Feature activation maps in different ways.

Image credit: the CUB200-2011 archive at https://www.vision.caltech.edu/datasets/cub_200_2011/, the NABirds archive at https://dl.allaboutbirds.org/nabirds, the VegFru archive at https://github.com/ustc-vim/vegfru.

These visualisations are denoted by the green box. In our proposed methodology, the initial branch makes use of the original features generated by the convolutional layer processing. On the other hand, the second branch incorporates features that are weighted by the attention of the ECA channel. During the fusion stage, we opt to forgo the cascade operation and employ a summation method to achieve the fusion of the two branches’ features. Based on the findings depicted in the figure, it is evident that our suggested approach effectively directs attention towards the target item, and the attention regions exhibit a wide range of diversity.

Convergence analysis

The mAP convergence curves of the Multi-FusNet on the CUB200-2011, FGVC Aircraft, and NABirds datasets are depicted in Fig. 11. These curves illustrate the variations in model performance throughout the training process. We conducted 300 iterations on three datasets, as depicted in the figure. Despite variations in dataset sizes, the approach consistently achieved convergence and stabilisation at the ideal mAP value within approximately 90 iterations. The experimental findings unequivocally illustrate the robust stability of our suggested approach.

Figure 11 Convergence curves of mAP values in some classic datasets.

Visualization of model retrieval effects

In this section, we present a visual representation of the retrieval outcomes achieved by our proposed approach on the five publicly available fine-grained image datasets that were discussed in ‘Datasets and evaluation metric’. In order to establish the validity of our studies, we performed two rounds of retrieval experiments on each dataset. In each experiment, the first column within each box denotes the input photos to be retrieved, while the second to sixth columns reflect the five query result images obtained as outcomes. In the conducted trials, the retrieval results were obtained by selecting the first ten most comparable photos. However, for the purpose of demonstrating the credibility of our approach in this particular section, we have chosen to display only the most representative five images. The results of this selection process are depicted in Fig. 12. The findings are displayed in Fig. 9. Based on the obtained results, it is evident that our approach demonstrates commendable retrieval performance across the four datasets, namely CUB-200-2011, Vegfru, Aircraft, and NABirds.

Figure 12 Retrieval example implemented according to the approach.

Image credit: the CUB200-2011 archive at https://www.vision.caltech.edu/datasets/cub_200_2011/, the FGVC-Aircraft archive at https://www.robots.ox.ac.uk/ vgg/data/fgvc-aircraft/, the Food101 archive at https://data.vision.ee.ethz.ch/cvl/datasets_extra/food-101/, the NABirds archive at https://dl.allaboutbirds.org/nabirds, the VegFru archive at https://github.com/ustc-vim/vegfru.

Discussion

In this study, we propose a novel method of multilevel feature fusion and self-attentive mapping coding for retrieving images of particular categories from a large number of fine-grained datasets. The results of our experiments indicate that the performance of image retrieval can be effectively enhanced by employing multilevel multidimensional features.

Numerous neural network-based fine-grained image retrieval studies (Xie et al., 2015b; Xu et al., 2015; Cai et al., 2023) have achieved tremendous success in recent years. These accomplishments are attributable to the rich feature extraction capability of deep neural networks. In order to further enhance the performance of image retrieval, the first stage is to optimise the performance of binary feature learning codes. However, the majority of existing deep hashing methods learn through high-level features; to address this issue, we are considering incorporating all middle- and low-level feature information. Due to its bottom-up and top-down feature mapping hierarchies that permit lateral connectivity of features at different scales (Liu et al., 2021; Wu et al., 2022), the feature pyramid approach, for example, has demonstrated outstanding ability to learn features for a wide variety of visual tasks. In contrast to previous studies that utilised altered features directly for visual task research, we use the final features for binary code mapping to aid in image retrieval.

In addition to mining various levels of information for image feature extraction using multilevel cascade networks, fusion of feature information utilising different dimensions is a popular area of research. Numerous studies have helped to enhance image retrieval performance through a fusion of features. Chen et al. (2023) proposed a semantic pyramid network for merging handcrafted features. Taheri, Rahbar & Beheshtifard (2023) retrieved remote sensing image data by fusing global and local features. Nevertheless, there are fewer studies that employ the aforementioned information in fine-grained images. Similar to previous research, we discovered that multi-feature fusion can enhance baselines by an average of 1 percent. In contrast to these studies, we only considered synthesis using information from the carefully enhanced image itself and did not employ multimodal data. Due to their plug-and-play and efficient qualities, attention mechanisms are extensively used in the field of vision tasks. In order to better extract fine-grained features for fusion, we compare the performance of the model after adding various attention modules, and find that adding the SE attention mechanism can further improve the retrieval accuracy. The results of our experiments indicate that image metadata can be better utilised by fusing augmented features at different levels, with the effect of fusing features augmented by channel weights being significantly better than that of spatially enhanced features; therefore, the potential of integrating spatially enhanced features should be explored further.

Overall, our proposed method for fine-grained image retrieval has numerous benefits and significantly enhances retrieval performance without requiring additional computation. This finding has significant implications for image retrieval using feature fusion with extended attentional focus.

Conclusions

In this research, we propose a multilevel feature fusion and self-attentive hash coding algorithm for large-scale fine-grained image retrieval tasks in this article.

Firstly, we design a multilevel feature mining network structure with a two-channel feature fusion module at each level, which facilitates the retention of multiscale and multidimensional image information. Subsequently, for enhance the contribution of key regions mapped to valid hash codes, we apply the self-attention structure for the first time to the fine-grained hash task. Finally, experiments on five publicly available fine-grained image datasets demonstrate the state-of-the-art of our method, generalisation experiments on five additional introduced image datasets demonstrate the generality of our approach, while exhaustive ablation experiments and a lots of comparison experiments with different modules validate our contribution. This work will encourage researchers to examine the potential of feature fusion and self-attention mechanisms in a variety of deep hashing applications.

In the future, we would like to investigate the contribution of self-attention to efficient hash code mapping through deep features in multiple information domains. This is in addition to focusing on other modal information alongside feature fusion using channel information, and even extending the focus beyond the RGB domain (e.g., the frequency domain). We believe that the widespread application of fine-grained image retrieval techniques will benefit the medical, industrial, and remote sensing fields, which heavily rely on the support of large amounts of data, and that this advancement will play a critical role in promoting technological advancements in related industries, thereby making the work of researchers and practitioners easier.

Additional Information and Declarations

Competing Interests

Author Contributions

Data Availability

The authors declare there are no competing interests.

Xiaohui Cui conceived and designed the experiments, prepared figures and/or tables, and approved the final draft.

Huan Li conceived and designed the experiments, performed the experiments, prepared figures and/or tables, authored or reviewed drafts of the article, and approved the final draft.

Lei Liu analyzed the data, prepared figures and/or tables, and approved the final draft.

Sheng Wang performed the computation work, authored or reviewed drafts of the article, and approved the final draft.

Fu Xu conceived and designed the experiments, authored or reviewed drafts of the article, and approved the final draft.

The following information was supplied regarding data availability:

The code is available at GitHub and Zenodo:

- https://github.com/BJFU-CS2012/MuiltNet

wyyl (2024). BJFU-CS2012/MuiltNet: v2.0.0 (v2.0). Zenodo. https://doi.org/10.5281/zenodo.10846524.

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
