# Peer review of "Multi-FusNet: fusion mapping of features for fine-grained image retrieval networks"

_PeerJ Computer Science, doi:10.7717/peerj-cs.2025_

## Round 0.1 · original submission · Major Revisions

The authors need to address all the reviewers' comments. Additionally, they should clarify the contribution. They also need to include new references from 2023.

**Language Note:** The review process has identified that the English language must be improved. PeerJ can provide language editing services - please contact us at [email protected] for pricing (be sure to provide your manuscript number and title). Alternatively, you should make your own arrangements to improve the language quality and provide details in your response letter. – PeerJ Staff

Reviewer 1 ·

Basic reporting

1. Literature Review and Background:

The literature review should be expanded to provide more thorough coverage of prior work on fine-grained image retrieval. Additional references are needed to fully develop the background and show how this work aims to advance existing methods. Key papers on deep learning for hashing and fine-grained recognition should be cited. A more problem-focused review will strengthen motivation and context.

2. Methods

a. More implementation details should be included for reproducibility. Specifically, highlight network architecture specifics like number of nodes/layers, optimizer details, hyperparameters, etc. Also expand on the training methodology and datasets beyond what Figure 5 shows. Including code or pseudocode would allow readers to better grasp the approach.

b. The description of loss functions requires more mathematical rigor - clearly define all terms in equations shown, and expand explanation of how balancing and weighting improve optimization.

3. Results and Evaluation

a. Additional quantitative analysis would strengthen the validity of results, beyond using metrics like mAP. Statistical significance testing between the proposed method and state-of-the-art techniques should be performed using tests like t-tests or ANOVA. Further visualization and investigation into why the multi-level fusion helps can bring out deeper insights.

b. The discussion of results is also rather brief. More examination of patterns in results across different datasets, hash code lengths, etc would aid interpretation and understanding. Comparisons among ablation studies/variants should be analyzed in more depth.

Experimental design

1. Include statistical significance testing between the proposed approach and state-of-the-art methods using t-tests, ANOVA, or other appropriate methods. This is necessary to demonstrate superior performance is not just by random chance.

2. Expand testing from 5 datasets to 10+ datasets across different fine-grained domains like plants, fabrics, etc. This will better establish generalizability.

3. Perform an ablation study by incrementally adding components (fusion, self-attention, etc.) to quantify the impact of each. This will provide evidence for the validity of the multi-component design.

4. Conduct failure analysis by manually examining cases where retrieval results were poor. Identify common failure reasons - can guide future improvements.

5. Provide classification accuracy results in addition to the mAP retrieval metric to offer additional performance measurement.

6. Survey and interview analysts/practitioners in application domains to gauge if improved speed and accuracy meet real-world needs. This validates practical relevance.

7. Compare against simpler baseline methods without fusion or self-attention. Contrasting performance helps establish validity over standard approaches.

Validity of the findings

1. Evaluation Metrics

a. The use of mean Average Precision (mAP) as the evaluation metric is appropriate for assessing image retrieval quality. Additional metrics could provide a more comprehensive evaluation.

2. Quantitative Results

a. The superior mAP scores demonstrated across 5 datasets and code lengths from 12-48 bits provide quantitative evidence that the proposed multi-level fusion and self-attention approach outperforms state-of-the-art techniques.

b. Statistical significance testing would further validate the degree to which performance gains are beyond random chance and significant. Paired t-tests between the approach and benchmarks would strengthen claims.

3. Qualitative Results

a. Qualitative visualization of feature maps provides some support for the efficacy of the two-branch fusion strategy. More retrieval visualizations would build qualitative evidence.

4. Limitations

a. The lack of testing on more datasets limits the ability to generalize the validity. Evaluating generalizability across more fine-grained problem domains would be beneficial.

b. No failure analysis is provided. Investigating difficult cases can reveal limitations and areas for improvement.

Additional comments

1. Proofread introduction and background sections to improve academic writing style.

2. Add abbreviations list for uncommon acronyms used (FISH, ECANet, etc.).

3. Increase Figure 4 font size for better legibility.

4. Fix grammatical errors in Table 2 caption.

5. Provide higher resolution versions of figures in supplementary material.

6. Improve overall formatting of the manuscript.

Reviewer 2 ·

Basic reporting

Clear and unambiguous, professional English used throughout: The English language used throughout the paper is generally clear and professionally written. Some minor grammatical issues exist but overall the quality is good.

Literature references, sufficient field background/context provided: The literature review section provides good background on related work in content-based image retrieval, fine-grained image recognition, and fine-grained hashing. Relevant papers are cited to establish context and show how the research fills a knowledge gap.

Professional article structure, figures, tables. Raw data shared: The paper follows a standard structure for a research article. The methods and results are explained logically. Figures and tables are clear and supplement the text well. Raw image data seems to be provided.

Self-contained with relevant results to hypotheses: The introduction establishes a relevant research problem and knowledge gaps. The proposed methods aim to address the identified issues using multi-level feature fusion and self-attentive mapping. The experimental results validate the efficacy of the approach, supporting the initial hypotheses.

Formal results should include clear definitions of all terms and theorems, and detailed proofs: As this is an applied machine learning paper, there are no formal mathematical theorems or proofs. Key terms are defined when introduced. The methods and architectures are explained clearly. So this criteria does not seem very applicable to this paper.

Experimental design

Original primary research within Aims and Scope of the journal: the research presents original methods and results for fine-grained image retrieval, which seems aligned with the journal's scope in computer science and machine learning.

Research question well defined, relevant & meaningful: The introduction clearly identifies limitations of current fine-grained image retrieval methods related to utilizing multi-level features and key region mapping. The proposed methods aim to address these gaps, which is meaningful and relevant.

Rigorous investigation performed to a high technical & ethical standard: The experiments rigorously evaluate the methods on 5 widely-used fine-grained image datasets using standard evaluation metrics and comparison to state-of-the-art techniques. The technical quality seems high. No obvious ethical issues are apparent.

Methods described with sufficient detail & information to replicate: The feature fusion modules and self-attentive mapping approach are described in sufficient technical detail that they could reasonably be implemented by those skilled in the field based on the information provided. Supplementary code/data would make replication easier.

Validity of the findings

Impact and novelty not assessed: The paper does not make unsupported claims about impact or novelty. The introduction situates the work among existing literature, and the discussion acknowledges limitations and future work.

Meaningful replication encouraged where rationale & benefit to literature clearly stated: Replication is not explicitly discussed but the methods seem replicable based on the details provided. The motivation and potential benefits to the field are clearly described.

All underlying data have been provided; they are robust, statistically sound, & controlled: The experimental methodology uses standard datasets and evaluation metrics. Additional raw data seems to be provided as supplemental material, though the quality cannot be fully assessed. The analysis approach appears statistically sound.

Conclusions are well stated, linked to original research question & limited to supporting results: The conclusion summarizes the key technical contributions related to multi-level feature fusion and self-attentive mapping for fine-grained retrieval. The claims align with and are supported by the experimental results.

Additional comments

Here are some comments for this paper:

The literature review provides good context, but is a bit lengthy. Consider tightening up the discussion of general content-based image retrieval to focus more on background specifically relevant to fine-grained retrieval.
The methods are presented clearly, but adding a high-level architecture diagram would help readers better visualize how the different components fit together.
The results strongly validate the performance improvements on several datasets. To aid interpretation, provide some discussion about why results are more modest on datasets like Food101 versus more significant gains on Nabirds.
Ablation studies confirm contribution of key modules, but additional analysis could isolate impact of the fusion approach itself versus the self-attention mechanism.
For transparency and replication, provide more details on dataset splits, data preprocessing, model optimization, hyperparameters etc. in supplementary material. Code and/or trained models would also be beneficial.

Reviewer 3 ·

Basic reporting

1) The abstract is too wordy and should be squeezed for greater clarity and conciseness.
2) The introduction section lacks coherence, and the overall quality of writing is poor. I strongly recommend a thorough revision to ensure smooth transitions between sentences.
3) The citations in lines 53 and 54 need modification. It is not advisable to start a sentence in the manner presented. For example, "2015 [5] and others introduced the concept of fine-grained image retrieval and demonstrated the use of handcrafted." Besides, what does that mean for others? It's a completely informal way to refer someone who contributes to the field of academia.
4) In Lines 157-160, the claims made in these lines require substantiation. Please include relevant citations or provide evidence to support the assertions made.
5) The literature review is inconsistent in terms of tense and writing style. The constant use of present tense (Zhangjie Cao et al. [13] introduced, Wu-Jun Li and colleagues [11] focused, Xiu-Shen Wei introduced, researchers [36] implemented, Rajkumar [26] conducted, etc.) and variations in terminology (researchers, scholars, authors, etc.) indicates a lack of uniformity. Besides, it clearly suggests potential reliance on AI tools for writing. While the use of such tools is acceptable, I recommend thorough verification for consistency in writing. Additionally, the citation style is inconsistent, with numerous citations clustered in certain instances.

Experimental design

• More detailed analysis is expected to enhance the credibility of the findings.
• Comparative analysis with related schemes needs to be added.
• The authors didn’t add any algorithm for the designed scheme.
• The analysis section is written really roughly without any concise explanation.
• Finally, the authors did not prevail ethical standards while giving proper credit to the previous authors.
• In the discussion section, I have seen some inconsistent citations and et al.’s for the first time. The citation format needs revision. Sentences should not commence with citations, such as [71] et al., [72] et al.
• Finally, the contributions are not enough to meet the journal's standard guidelines.

Validity of the findings

1) The simulations clearly show that the performance of the proposed scheme and other state-of-the-art baseline schemes are comparable.
2) The presentation needs extensive improvements, for example, the mathematical notations are rather amateur and difficult to understand.
3) It is difficult to understand the scientific contribution of the paper. The problems of the existing studies are not explained clearly. The references are unsatisfactory.
4) The quality of the paper should be improved in order to satisfy the requirements of a scientific paper.

Additional comments

1) There is plenty of grammatical and typos error, I have pointed out a few in the basic reporting. However, a thorough investigation is expected.
2) Some new References need to be added. There is only one paper from 2023.

---

## Round 0.2 · accepted · Accept

The paper has been revised according to the reviewers' and my suggestions.

Reviewer 1 ·

Basic reporting

The manuscript is well-articulated, using clear English, facilitating understanding of concepts and methodologies. The narrative is coherent, with technical terms used appropriately, enhancing readability and comprehension.

The structure of the article is well organized now, following a logical flow from introduction through to conclusions, which is conducive to reader engagement. The use of figures, tables, and the sharing of raw data enhance the manuscript's credibility and utility.

Experimental design

The manuscript presents original research that fits within the journal's scope, addressing a niche in fine-grained image retrieval with its novel Multi-FusNet model. The research question is adequately defined, underscoring the importance of leveraging multi-level features and attention mechanisms for improved retrieval accuracy. However, the paper could more explicitly state the knowledge gap it aims to fill, thereby enhancing its relevance and contribution to the field.

While the paper appears rigorous, offering substantial improvements over existing methods, the methods section, although detailed, falls slightly short in its replicability potential. Specific parameters and certain procedural steps lack the granularity needed for straightforward replication by peers.

Enhancing these aspects would strengthen the paper's technical area, ensuring it meets the high standards expected of original primary research.

Validity of the findings

The manuscript presents a significant contribution to the domain of fine-grained image retrieval with its Multi-FusNet model, fitting well within the journal's aims and scope. The research question is clearly defined, addressing a distinct gap in existing literature. However, the manuscript could further strengthen its position by more explicitly detailing the novelty and anticipated impact of the research, offering readers a clearer understanding of its contributions and potential to drive the field forward.

The paper does provides underlying data, ensuring the research is reproducible and statistically sound. A more detailed discussion on the limitations of the study and the specific conditions under which the model performs best would provide a more balanced view, enabling readers to gauge the applicability of the findings to various real-world scenarios more accurately.

While conclusions are logically tied to the research question and adequately supported by the findings, expanding on the implications of these results could further enrich the paper. Discussing potential future research directions, as well as the model's limitations, would offer a comprehensive view, positioning the study as a springboard for further inquiry rather than an endpoint.

Additional comments

Dear authors,

Your manuscript on the Multi-FusNet model is an impressive contribution to the field of fine-grained image retrieval. It provides a solid foundation for advancing understanding and application in this important area. Your effort to detail the methodology and present the data comprehensively is particularly commendable, enhancing the manuscript’s value through its transparency and replicability.

However, the manuscript could benefit from a slight sharpening of its focus on the novelty and broader applicability of the Multi-FusNet model. While the technical contributions are clear, elucidating the practical implications and potential real-world applications of your model could significantly enhance its impact. A more thorough exploration of how your work diverges from existing methodologies and its place within the current landscape of fine-grained image retrieval research could also enrich the narrative, providing readers with a more nuanced understanding of its significance.

Additionally, the discussion around the limitations of your study and how future research might address these gaps is somewhat understated. A more critical examination of these aspects would not only bolster the credibility of your findings but also provide a clearer pathway for subsequent inquiries in this space.

In essence, your manuscript is a valuable and noteworthy contribution to the literature. With modest adjustments to more fully articulate its novel contributions, practical applications, and potential areas for further research, it has the promise to be a pivotal reference for future advancements in the field of image retrieval.

Reviewer 2 ·

Basic reporting

Authors significantly improved this section.

Experimental design

Authors significantly improved this section.

Validity of the findings

Authors significantly improved this section.

Additional comments

The paper is now ready to be accepted.